**Data Availability Statement:** All relevant data are within the paper and its Supporting information files.

**Funding:** This work was supported by the Program of the Network-Type Joint Usage/Research Center

# Genomic analysis for the prediction of prognosis in small-bowel cancer

Akiyoshi Tsuboi[1◯], Yuji Urabe [2◯], Shiro Oka [1]*, Akihiko Sumioka[1], Sumio Iio[1], Ryo Yuge[3], Ryohei Hayashi[3], Toshio Kuwai[4], Yasuhiko Kitadai[5], Kazuya Kuraoka[6], Koji Arihiro[7], Shinji Tanaka[3], Kazuaki Chayama[1,8,9]

1 Department of Gastroenterology and Metabolism, Hiroshima University Hospital, Hiroshima, Japan, 2 Division of Regeneration and Medicine Center for Translational and Clinical Research, Hiroshima University Hospital, Hiroshima, Japan, 3 Department of Endoscopy, Hiroshima University Hospital, Hiroshima, Japan, 4 Department of Gastroenterology, National Hospital Organization, Kure Medical Center and Chugoku Cancer Center, Kure, Japan, 5 Department of Health and Science, Prefectural University of Hiroshima, Hiroshima, Japan, 6 Department of Diagnostic Pathology, National Hospital Organization, Kure Medical Center and Chugoku Cancer Center, Kure, Japan, 7 Department of Anatomical Pathology, Hiroshima University Hospital, Hiroshima, Japan, 8 Research Center for Hepatology and Gastroenterology, Hiroshima University, Hiroshima, Japan, 9 RIKEN Center for Integrative Medical Sciences, Yokohama, Japan

◯ These authors contributed equally to this work.
* oka4683@hiroshima-u.ac.jp

## Abstract

The current understanding of clinicopathological features and genomic variants of small-bowel cancer is limited, in part due to the rarity of the disease. However, understanding of these factors is necessary for the development of novel therapeutic agents for small-bowel cancer. Thus, we aimed to identify the clinicopathological features and genomic variants associated with its prognosis and recurrence. We retrospectively examined 24 consecutive patients with primary small-bowel cancer surgically treated between May 2005 and August 2018 and collected 29 tumor specimens. The 29 lesions were subjected to mismatch repair status evaluation, using immunohistochemistry (IHC), and targeted genomic sequencing, after which they were analyzed using a panel of 90 cancer-related genes. IHC revealed that 45% (13/29) of the lesions exhibited deficient mismatch repair. The most common genomic variants in small-bowel cancers were in *TP53* (48%, 13/27), followed by *KRAS* (44%, 12/27), *ARID1A* (33%, 9/27), *PIK3CA* (26%, 7/27), *APC* (26%, 7/27), and *SMAD4*, *NOTCH3*, *CREBBP*, *PTCH1*, and *EP300* (22%, 6/27 each). Overall survival and disease-specific survival of patients with tumor mutational burden (TMB) ≥10 mutations/Mb (n = 17) were significantly better than those of patients with TMB <10 mutations/Mb (n = 6). Additionally, patients with a mutant *SMAD4* had poorer recurrence-free survival than those with wild-type *SMAD4*. Our results suggested that TMB and *SMAD4* mutations were associated with the prognosis of small-bowel cancer patients. Thus, cancer genomic analysis could be useful in the search for biomarkers of prognosis prediction in small-bowel cancers.

for Radiation Disaster Medical Science of Hiroshima University, Nagasaki University, and Fukushima Medical University. The funder had no role in study design, data collection and analysis, decision to publish, or preparation of the manuscript.

**Competing interests:** The authors have declared that no competing interests exist.

## Introduction

Although the small-bowel constitutes three-quarters of the entire digestive tract, small-bowel cancer is rare compared to other gastrointestinal cancers. Small-bowel cancers have been reported to account for only approximately 3% of all gastrointestinal tract tumors [1]. However, the incidence of small-bowel cancer has increased in Western countries over the past several decades [2].

Raghav et al. [3] reported that compared to those with colorectal cancer (CRC), patients with small-bowel cancer are younger, and a higher proportion of men than women are affected by this condition. The risk factors of small-bowel cancer are unclear; however, predisposing factors are known to include hereditary syndromes such as familial adenomatous polyposis (FAP), Peutz-Jeghers syndrome (PJS), Lynch syndrome (LS), and inflammatory bowel diseases (IBDs) such as Crohn's disease and celiac disease, and obesity.

Although FAP, LS, Crohn's disease, and celiac disease are the predisposing factors of small-bowel cancer in the USA and Europe, a recent Japanese multicenter study reported these factors as not associated with the development of small-bowel cancer [4]. The prevalence of celiac disease in Caucasians is as high as 1% of the population, and it has been increasing [5]. Fukunaga et al. [6] reported the prevalence of celiac disease being 0.05% in a non-patient Japanese population. Therefore, the prevalence of small-bowel cancer associated with celiac disease is considered to be lower in Japan than that in the Western countries. Moreover, in a previous report on racial disparity with respect to gastrointestinal cancer, the incidence of small-bowel cancer was reported to vary among different races [7]. If race affects carcinogenic risk, then genomic variants may exist among different races.

Exploring the genetic landscape of cancers of various organs has recently become feasible with the growing availability of next-generation sequencing (NGS), regardless of the stage of cancer [8–10]. Several reports have been published on the genomic landscape of small-bowel cancer [11–15]. However, disease structure is different between Caucasian and Japanese populations. For example, background factors such as the frequency of celiac disease, Crohn's disease, and obesity are different between Japan and Western countries. The frequency of obesity varies across countries. According to World Health Organization (2016), a 20–40% frequency of body mass index (BMI) >30 was reported in Western countries, whereas a 4.4% frequency of BMI >30 was reported in Japan; this scenario is explained by Japanese people tending to consume diets low in fat, sugar and fructose. These factors for small-bowel cancer are likely to be impacted by genomic variants. For the development of novel therapeutic agents, it is necessary to clarify the characterization of clinicopathological features and genomic variants associated with carcinogenesis of small-bowel cancer. In this study, we aimed to clarify the genomic landscape for small-bowel cancer and association between clinicopathological features and genomic variants in the Japanese population. Our study revealed the MMR status and genetic variants of small-bowel cancer in a Japanese population.

## Methods

### Patients

We retrospectively analyzed 29 small-bowel cancer lesions in a total of 24 consecutive patients surgically treated at two hospitals in Japan from May 2005 to August 2018. Patients with FAP and duodenal cancer were excluded from the study. Data for each patient were obtained through a retrospective medical record review and from stored endoscopic findings. Tissue samples were collected from those archived in the pathology laboratories of Hiroshima University Hospital and National Hospital Organization Kure Medical Center and Chugoku

Cancer Center, Japan. The study was performed in accordance with the ethical standards of the Declaration of Helsinki. The study design and protocol were approved by the Independent Ethics Committee of Hiroshima University (approval number: E-1407; registration date: October 26, 2018) and National Hospital Organization Kure Medical Center and Chugoku Cancer Center (approval number: 2019–10; registration date: May 31, 2019). Written informed consent was obtained from all patients for the use of medical records and tissue samples. Regarding informed consent, we provided patients the opportunity to opt-out of the study by posting and presenting details of the study on our website. None of the patients refused inclusion during the study period.

## Histopathological findings

The resected specimens were fixed in 10% buffered formalin solution and embedded in paraffin. The archived paraffin-embedded samples were sliced into 2–3-μm-thick sections and stained with hematoxylin and eosin. Histological type and depth of tumor (T classification), lymph node metastasis (N classification), metastasis to other organs (M classification), and pathological staging were categorized in accordance with the Japanese Classification of Colorectal Carcinoma [16].

## Immunohistochemistry

Paraffin-embedded human small-bowel cancer tissue specimens were cut into 2–3-μm thick sections and mounted on positively charged slides. Antigen retrieval was performed using Tris-EDTA buffer (pH 9.0) in a microwave oven at 800 W for 5 min and at 150 W for 10 min. The tissues on the slides were then incubated with the following primary antibodies: monoclonal mouse anti-human MLH1 antibody (BD Biosciences, San Jose, CA, USA; dilution, 1:10; 2 h), anti-human MSH2 antibody (33–7900; Invitrogen, Carlsbad, CA, USA; dilution 1:200; 2 h), anti-human MSH6 antibody (12988; Cell Signaling Technology, Danvers, MA, USA; dilution, 1:500; 2 h), and anti-human PMS2 antibody (B-NU844UCE; BioGenex, Fremont, CA, USA; undiluted; 2 h). All incubations were performed at room temperature (20–25 ˚C). The bound primary antibodies were detected using the Dako EnVision+ System (Dako, Copenhagen, Denmark). After incubation for 1 h at room temperature with corresponding peroxidase-conjugated secondary antibodies, a positive signal was detected by exposure to stable 3,3′-diaminobenzidine for 5 to 10 min. The immunostained tissue sections were counterstained with hematoxylin. All immunohistochemical assessments were conducted blinded without the evaluator having any knowledge of histological diagnoses. Tumor proteins were classified as mismatch repair (MMR)-deficient (dMMR) or MMR-proficient (pMMR).

## DNA extraction

DNA was extracted from 40 μm of formalin-fixed, paraffin-embedded (FFPE) sections. The sections were subjected to hematoxylin-eosin review to ensure that a minimum of 60% of the DNA would be derived from tumor cells. DNA was extracted from the tissues using a GeneRead DNA FFPE Kit (Qiagen, Hilden, Germany) according to the protocol of the manufacturer. The extracted DNA was eluted into 40 μL of Elution buffer, quantified using Qubit dsDNA HS (high sensitivity) Assay Kit (Thermo Fisher Scientific, Waltham, MA, USA), and stored at 4˚C until use. The FFPE-derived DNA samples were quantified by calculating the normalized DNA integrity scores (ΔΔCq) obtained by quantitative polymerase chain reaction (qPCR) according to the Agilent NGS FFPE QC Kit (Agilent Technologies, Santa Clara, CA, USA) protocol.

## Target enrichment and NGS

DNA extracted from tumors and normal small-bowel mucosa was fragmented into 150–200 bp-long fragments with a restriction enzyme using SureSelect XT HS Reagent Kit and SureSelect XT Low Input Enzymatic Fragmentation Kit (Agilent Technologies, Santa Clara, CA, USA). The fragments were used for library construction according to the SureSelect XT HS Reagent Kit protocol. Exons of 90 oncogenes and the associated introns of 35 fusion oncogenes were enriched using the SureSelect NCC Oncopanel (Agilent Technologies). The oncogenes containing the 90 target exons are listed in S1 Table. The resulting pooled libraries were checked for quality control using an Agilent High Sensitivity D1000 Screen Tape System with the 2200 TapeStation Instrument (Agilent Technologies). Sequencing and paired-end reads were performed using the HiSeq X platform (Illumina, San Diego, CA, USA). Tumor mutational burden (TMB) was calculated as the number of somatic base substitutions or insertions/ deletions (indels) per megabase (Mb) of coding DNA in the target region of the test after filtering to remove the known somatic and deleterious mutations and extrapolating that value to the exome or genome as a whole.

## Variant detection

Sequencing reads were aligned to the hg38 reference sequence and analyzed using SureCall Software version 4.1 (Agilent Technologies). To improve the mapping quality, PCR duplicates were removed prior to variant calling based on molecular barcodes using the SureCall Software. Paired-end and single analysis as part of the SureCall Software was used to identify single nucleotide variants and indels in tumors. Called variants were considered germline mutations if they were also present in the normal small-bowel mucosa tissue. To reduce the false-positive rate, cutoff values for somatic mutations in tumors were set as a read depth >20 and forward/ reverse balance between 0.25 and 0.75. The SureCall SNP caller was configured using the Sure-Select default settings of variant score threshold at 0.3, minimum quality for base at 30, variant call quality threshold at 100, minimum allele frequency at 0.05, and the minimum number of reads supporting variant allele at 10. Moreover, variants were excluded as somatic mutation candidates in all sample types if they were (a) repeated sequences registered in the University of California Santa Cruz repeat masker, (b) called as replacements, or (c) clearly identified as sequence errors in the Integrated Genomic Viewer (Broad Institute).

## Patient evaluation

We evaluated clinicopathological features including patient age, sex, BMI, chief complaint, past history of other organ cancers, predisposing conditions, familial history, number of lesions, carcinoembryonic antigen (CEA) and carbohydrate antigen 19–9 (CA19-9) values, diagnostic modality, tumor location, tumor morphology, depth of tumor invasion (T), lymph node metastasis (N), distant metastasis (M), pathological staging according to the Japanese Classification of Colorectal Carcinoma [16], and location of distant metastasis. Furthermore, MMR status based on immunohistochemistry and small-bowel cancer genomic variants were evaluated. Finally, MMR status, TMB, and genomic variants of *KRAS*, *TP53*, *PIK3CA*, and *SMAD4* associated with prognosis after surgery and recurrence after R0 resection were analyzed.

## Statistical analyses

Continuous variables are presented as mean ± standard deviation and were compared by Student's *t*-test. Dichotomous variables were compared by Fisher's exact test. Overall survival

(OS) was defined as the time from surgery to death due to any cause. Disease-specific survival (DSS) was defined as the time from surgery to death caused by small-bowel cancer. Recurrence-free survival (RFS) was defined as the time from R0 surgery to first recurrence (local or distant). OS, DSS, and RFS were estimated using the Kaplan-Meier method. Survival curves were compared using the log-rank test. To assess the association among patient characteristics, genomic variants, and survival, Cox proportional hazards models were used, and multivariable regression analyses were fitted by stepwise selection methods. All statistical analyses were performed using the JMP statistical software version 13 (SAS Institute Inc., Cary, North Carolina, USA), and *P* values less than 0.05 were considered significant.

## Results

Baseline characteristics of 29 small-bowel cancers of 24 patients are shown in Tables 1 and 2. Two of the 24 patients were diagnosed with LS and had cancers in other organs (one had gastric cancer and the other had CRC).

Furthermore, we evaluated the MMR status in the 29 small-bowel cancer lesions resected by surgery (S2 Table). IHC revealed 45% (13/29) of the lesions to be dMMR with MLH1 and PMS2 being absent in 7 lesions, MSH2 and MSH6 being absent in 3 lesions, and MSH6 being absent in 3 lesions. Based on the Kaplan-Meier survival analysis, there were no differences between dMMR and pMMR with respect to OS or DSS (S1 Fig).

Of the 29 resected lesions from the 24 patients with small-bowel cancer, 2 samples were excluded from genetic analysis as the amount of DNA was lower than 100 ng. Genomic variant results from the examination of the remaining 27 small-bowel cancer lesions, from 23 patients, are shown in Fig 1. The most common genomic variants were in *TP53* (48%, 13/27), followed by *KRAS* (44%, 12/27), *ARID1A* (33%, 9/27), *PIK3CA* (26%, 7/27), *APC* (26%, 7/27), and *SMAD4*, *NOTCH3*, *CREBBP*, *PTCH1*, and *EP300* (22%, 6/27 each).

The median TMB value was calculated to be 14 mutations/Mb. There were 9 lesions with TMB <10 mutations/Mb and 18 lesions with TMB ≥10 mutations/Mb. There were no differences in clinicopathological features between the lesions with TMB ≥10 mutations/Mb and those with TMB <10 mutations/Mb (S3 Table). All lesions with TMB ≥10 mutations/Mb were depressed type morphology with the tumor morphology classified as Type 2 or Type 3. In comparison, 33% (3/9) of the lesions were protruded type morphology in the lesions with TMB <10 mutations/Mb. The frequency of the depressed type morphology in the lesions with TMB ≥10 mutations/Mb was significantly higher than that of protruded type (*P* = 0.03). IHC results for lesions with TMB ≥10 mutations/Mb and TMB <10 mutations/Mb are shown in Table 3. In lesions with TMB ≥10 mutations/Mb (n = 18), the frequency of dMMR based on IHC was 56% (10/18). In contrast, the frequency of dMMR based on IHC in lesions with TMB <10 mutations/Mb was 11% (1/9).

Associations between prognosis and genomic variant based on Kaplan–Meier analysis are shown in S2 and S3 Figs. There were no differences in the rates of OS and DSS between genomic mutations of each gene; however, the 5-year OS and DSS were 55% and 58% for TMB ≥10 mutations/Mb and 33% and 33% for TMB <10 mutations/Mb. Thus, our results showed that TMB was significantly associated with worse prognosis [TMB <10 mutations/Mb (n = 6) vs. TMB ≥10 mutations/Mb (n = 17), log-rank *P* < 0.05] (Fig 2). The other covariates from the Cox proportional hazard analysis associated with OS are shown in Tables 4 and 5. Regarding the univariate analysis, pathological Stage IV (HR, 12.13; 95% CI, 2.89–83.36; *P* < 0.01) and R1/2 resection (HR, 7.7; 95% CI, 2.06–37.25; *P* < 0.01) were associated with OS (Table 4), whereas TMB <10 mutations/Mb was not significantly correlated with OS (HR, 3.29; 95% CI, 0.84–11.24; *P* = 0.08). However, in the multivariate analysis, pathological Stage IV (HR, 58.68;

**Table 1. Baseline characteristics of small-bowel cancer.** Characteristics patients with small-bowel cancer in our study.

| Variables | Small-bowel cancer (n = 24) |
|---|---|
| Age (years), mean ± SD | 61.7 ± 11.7 |
| Sex, male/female | 16/8 |
| BMI, mean ± SD | 22.3 ± 6.3 |
| Observation period (month), mean ± SD | 50.0 ± 43.3 |
| Chief complaint, +/− | 22/2 |
| Intestinal obstruction | 9 (38) |
| Abdominal pain | 8 (33) |
| Gastrointestinal bleeding | 5 (21) |
| History of other organ cancers | 7 (29) |
| Colorectal cancer | 3 (13) |
| Breast cancer | 2 (8) |
| Gastric cancer | 1 (4) |
| Lung cancer | 1 (4) |
| Bladder cancer | 1 (4) |
| Pancreatic cancer | 1 (4) |
| Endometrial cancer | 1 (4) |
| Gallbladder cancer | 1 (4) |
| Predisposing conditions | |
| FAP | 0 (0) |
| PJS | 0 (0) |
| LS | 2 (8) |
| Crohn's disease | 0 (0) |
| Celiac disease | 0 (0) |
| Number of lesions, single/ multiple | 21/3 |
| CEA (ng/mL), median [IQR] | 2.2 [1.4–5.9] |
| CA19-9 (U/mL), median [IQR] | 8.5 [2.0–77.9] |
| Diagnostic modality | |
| DBE | 20 (83) |
| CT | 3 (13) |
| CE | 1 (4) |
| Postoperative chemotherapy | |
| S1 | 5 (21) |
| FOLFOX | 4 (17) |
| CapeOX | 3 (13) |
| Capecitabine | 2 (8) |
| UFT/LV | 2 (8) |
| FOLFOX+Bmab | 1 (4) |
| S1+Doc | 1 (4) |
| None | 6 (25) |

Data represented as n (%) and mean ± SD.

SD: standard deviation, BMI: body mass index, FAP: familial adenomatous polyposis, PJS: Peutz-Jeghers syndrome, LS: Lynch syndrome, CEA: carcinoembryonic antigen, CA19-19: carbohydrate antigen 19–9, IQR: interquartile range, DBE: double balloon endoscopy, CT: computed tomography, CE: capsule endoscopy, S1: tegafur, gimeracil and oteracil, FOLFOX: fluorouracil, leucovorin and oxaliplatin, CapeOX: capesitabine and oxaliplatin, UFT/LV: uracil-tegafur and leucovorin, Bmab: bevacizumab, Doc: docetaxel.

**Table 2. Baseline characteristics of small-bowel cancer.** Characteristics of small-bowel cancer lesions in our study.

| Variables | Small-bowel cancer (n = 29) |
|---|---|
| Location | |
| Jejunum | 24 (83) |
| Ileum | 5 (17) |
| Endoscopic stricture | 11 (38) |
| Tumor diameter (mm), mean ± SD | 42.2 ± 18.4 |
| Histology | |
| tub/pap | 24 (83) |
| por/sig/muc | 5 (17) |
| Tumor morphology | |
| Type 1 | 3 (10) |
| Type 2 | 17 (59) |
| Type 3 | 9 (31) |
| TNM classification | |
| T0-2/T3–4 | 1/28 |
| N0–1/N2–3 | 23/6 |
| M0/M1 | 20/9 |
| Location of distant metastasis | |
| Peritoneum dissemination | 8 (28) |
| Liver | 2 (7) |
| Pathological stage | |
| Stage I | 1 (3) |
| Stage II | 9 (31) |
| Stage III | 10 (35) |
| Stage IV | 9 (31) |

Data represented as n (%) and mean ± SD.

SD: standard deviation, tub: tubular adenocarcinoma, pap: papillary adenocarcinoma, por: poorly differentiated adenocarcinoma, sig: signet-ring cell carcinoma, muc: mucinous adenocarcinoma.

95% CI, 7.89–1348.14; $P < 0.01$) and TMB <10 mutations/Mb (HR, 11.33; 95% CI, 2.08–85.05; $P < 0.01$) were associated with OS.

Meanwhile, the examination of 16 patients with specimens obtained through R0 resection for a correlation between genomic variant and recurrence revealed that the 5-year RFS were 65% for wild-type *SMAD4* and 20% for mutant *SMAD4*. Therefore, the patients with mutant *SMAD4* had significantly worse RFS than those with wild-type *SMAD4* (log-rank $P < 0.05$; Fig 3). In contrast, unlike OS and DSS, there was no association between TMB and RFS (S2 and S3 Figs). The other covariates from the Cox proportional hazard analysis associated with RFS are shown in Tables 6 and 7. Regarding the univariate analysis, pathological Stage IV (HR, 11.43; 95% CI, 1.09–247.04; $P = 0.04$) was correlated with RFS (Table 6), whereas mutated *SMAD4* was not significantly correlated with RFS (HR, 4.15; 95% CI, 0.90–21.41; $P = 0.07$). However, pathological Stage IV (HR, 24.07; 95% CI, 1.84–654.62; $P = 0.02$) and mutated *SMAD4* (HR, 6.72; 95% CI, 1.28–49.54; $P = 0.03$) were correlated with RFS in the multivariate analysis. Among all resection patients, 5 patients had mutated *SMAD4* gene, of which 3 patients were Stage II, 1 patient was Stage III, and 1 patient was Stage IV. Except for 1 patient of Stage II, all patients had recurrence of peritoneal dissemination. Of the 4 patients that had recurrence of small-bowel cancer, 2 patients died from primary cancer, whereas the other 2 patients are still alive and undergoing treatment with multidisciplinary treatment including additional surgery.

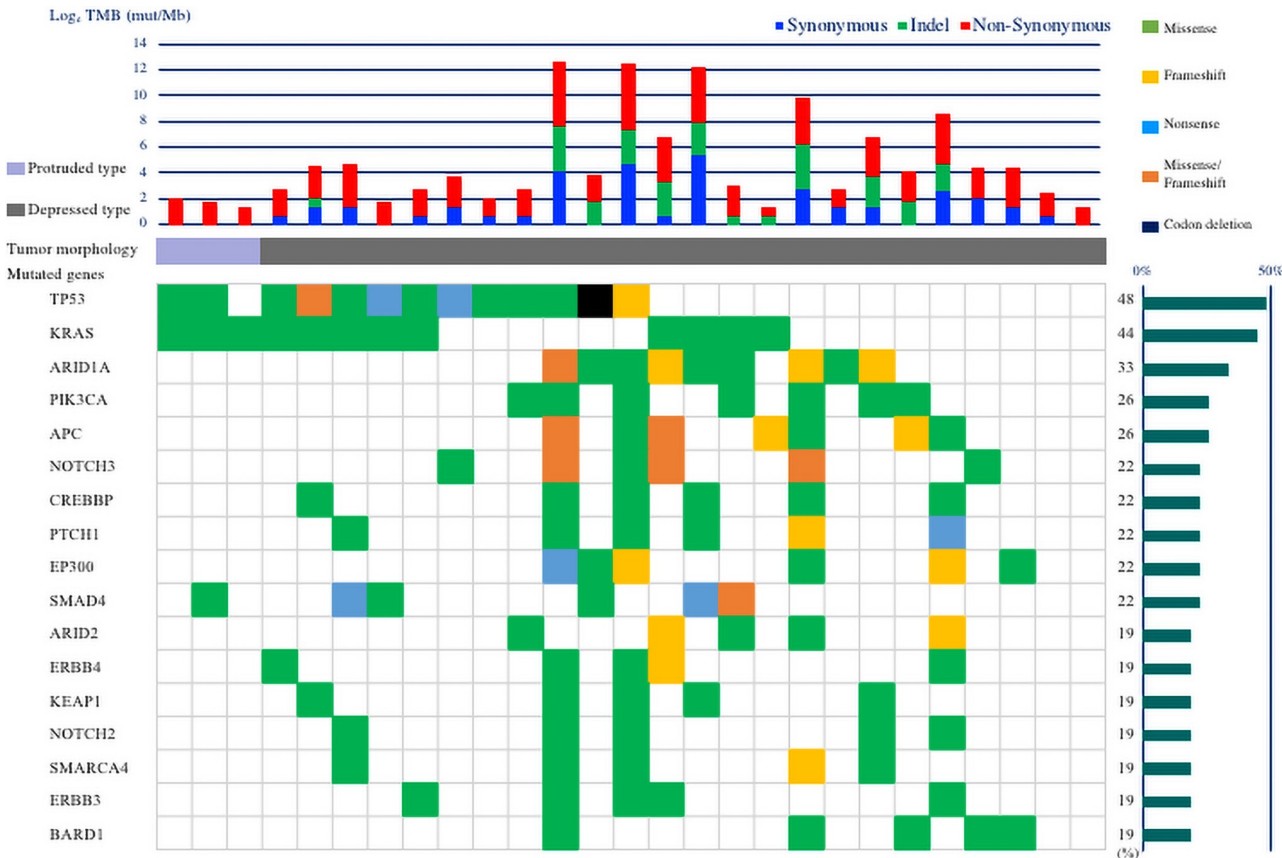

**Fig 1. Genomic landscapes of 27 small-bowel cancers. Upper panel**: Number of somatic mutations per sample. Blue bars indicate the number of synonymous mutations, green bars indicate the number of insertions/deletions (indels), and red bars indicate the number of non-synonymous mutations. **Middle panel**: Morphology of lesions. Light purple bars indicate protruded type tumor morphology (defined as Type 1 morphology) and gray bars indicate depressed type tumor morphology (defined as Type 2 and Type 3 morphology). **Lower panel**: Mutation pattern of 17 mutated genes with a frequency >15% from 27 small-bowel cancers. Mutated genes are shown. Green cells indicate missense mutations, yellow cells indicate frameshift mutations, light blue cells indicate nonsense mutations, orange cells indicate missense/frameshift mutations, and black cells indicate codon deletions. The bar graph on the right indicates the frequency of each mutated gene of small-bowel cancers.

**Table 3. Outcome of immunohistochemistry analysis of small-bowel cancer lesions with high TMB and low TMB.**

| Variables | TMB <10 mut/Mb (n = 9) | TMB ≥10 mut/Mb (n = 18) | *P*-value |
|---|---|---|---|
| MMR status | | | |
| Absent MLH1 and PMS2 | 0 (0) | 5 (28) | 0.14 |
| Absent PMS2 | 0 (0) | 0 (0) | 1.00 |
| Absent MSH2 and MSH6 | 1 (11) | 2 (11) | 1.00 |
| Absent MSH6 | 0 (0) | 3 (17) | 0.53 |
| dMMR | 1 (11) | 10 (56) | 0.04 |

Data represented as n (%).

TMB: tumor mutational burden, mut/MB: mutations per megabase of DNA, MMR: mismatch repair, dMMR: deficient mismatch repair.

(a)

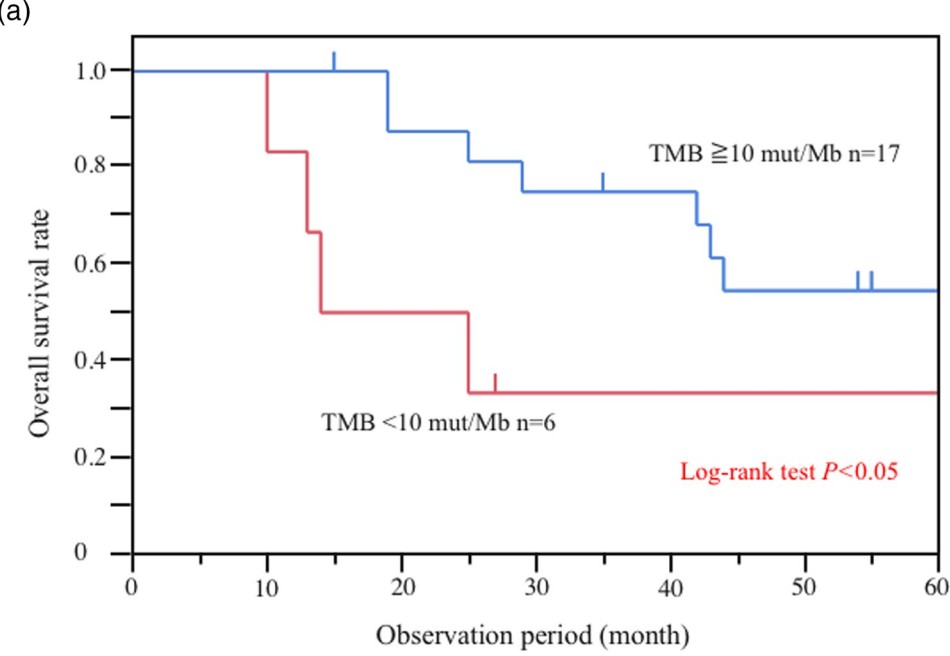

(b)

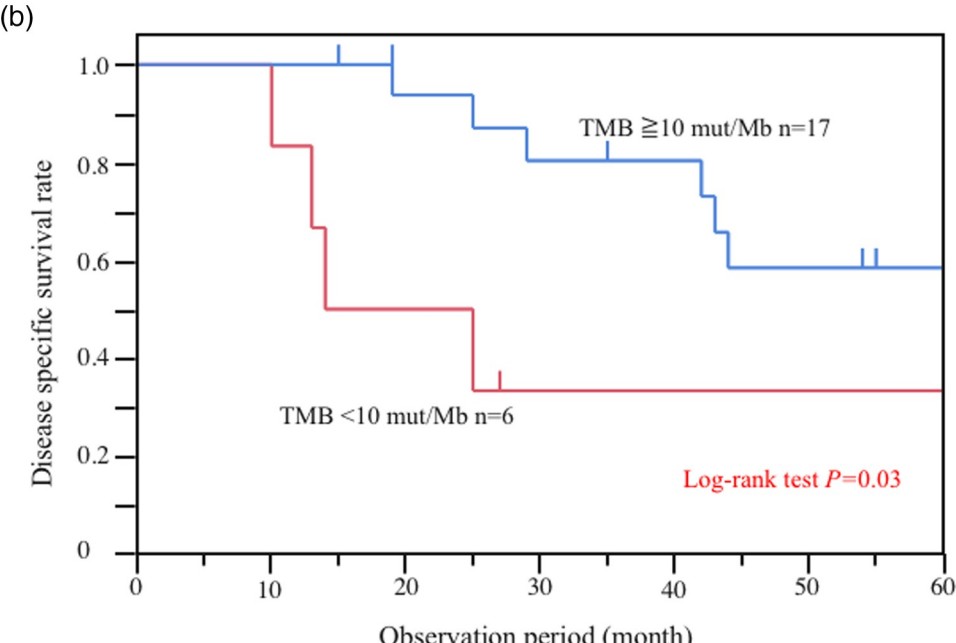

**Fig 2. Analysis of Overall Survival (OS) and Disease-Specific Survival (DSS) for patients with small-bowel cancer according to Tumor Mutational Burden (TMB).** (a) OS of patients with small-bowel cancer according to TMB [TMB <10 mutations/megabyte of DNA (mut/Mb) vs TMB ≥10 mut/Mb, $P < 0.05$]. (b) DSS of patients with small-bowel cancer according to TMB (TMB <10 mut/Mb vs TMB ≥10 mut/Mb, $P = 0.03$).

**Table 4. Univariate association with overall survival.**

| Variables | | N | OS | | |
|---|---|---|---|---|---|
| | | | HR (95%CI) | P-value | Log-rank P-value |
| Sex | Male | 16 | 2.60 (0.68–16.94) | 0.17 | 0.20 |
| | Female | 8 | Ref. | | |
| Age | >65 years | 8 | 0.86 (0.23–2.73) | 0.80 | 0.80 |
| | <65 years | 16 | Ref. | | |
| Stage | IV | 9 | 12.13 (2.89–83.36) | <0.01 | <0.01 |
| | I–III | 15 | Ref. | | |
| R0 resection | R1/2 | 8 | 7.7 (2.06–37.25) | <0.01 | <0.01 |
| | R0 | 16 | Ref. | | |
| Postoperative chemotherapy | Done | 18 | 2.47 (0.64–16.16) | 0.20 | 0.23 |
| | None | 6 | Ref. | | |
| IHC | dMMR | 11 | 1.25 (0.38–3.99) | 0.71 | 0.71 |
| | pMMR | 13 | Ref. | | |
| TMB | <10 mut/Mb | 6 | 3.29 (0.84–11.24) | 0.08 | <0.05 |
| | ≥10 mut/Mb | 17 | Ref. | | |
| *SMAD4* | Mutation | 6 | 0.52 (0.08–2.01) | 0.37 | 0.39 |
| | Wild | 17 | Ref. | | |

IHC: immunohistochemical, TMB: tumor mutational burden, mut/MB: mutations per megabase of DNA, MMR: mismatch repair, dMMR: deficient mismatch repair, pMMR: proficient mismatch repair, HR: hazard ratio, OS: overall survival.

Of the 2 patients that had recurrence at Stage II, one of them received postoperative adjuvant chemotherapy with capecitabine, and the other did not receive adjuvant chemotherapy.

## Discussion

Our study revealed the MMR status and genetic variants of small-bowel cancer in a Japanese population. We excluded patients of duodenal cancer in this study as there may be large differences in genomic variants between duodenal and jejunal/ileal cancers. In small-bowel cancers, duodenum is the most common tumor location, and yet, patients with duodenal cancer are known to be younger and are commonly diagnosed with a lower stage cancer than those with small-bowel cancer [17]. In addition, a previous population-based study reported that patients with small-bowel cancer have better prognosis than those with duodenal cancer [17]. Moreover, duodenum localization is a negative predictor of survival after resection of small-bowel cancer [17]. These results are likely to reflect the anatomic complexity of the retroperitoneally located duodenum. However, a Japanese study reported that 42% of the patients with duodenal cancer were diagnosed with early-stage disease (Stage 0/I) by screening via

**Table 5. Multivariate association with overall survival.**

| Variables | | N | OS | |
|---|---|---|---|---|
| | | | HR (95%CI) | P-value |
| Stage | IV | 9 | 58.68 (7.89–1348.14) | <0.01 |
| | I–III | 15 | Ref. | |
| TMB | <10 mut/Mb | 6 | 11.33 (2.08–85.05) | <0.01 |
| | ≥10 mut/Mb | 17 | Ref. | |

TMB: tumor mutational burden, mut/MB: mutations per megabase of DNA, HR: hazard ratio, OS: overall survival.

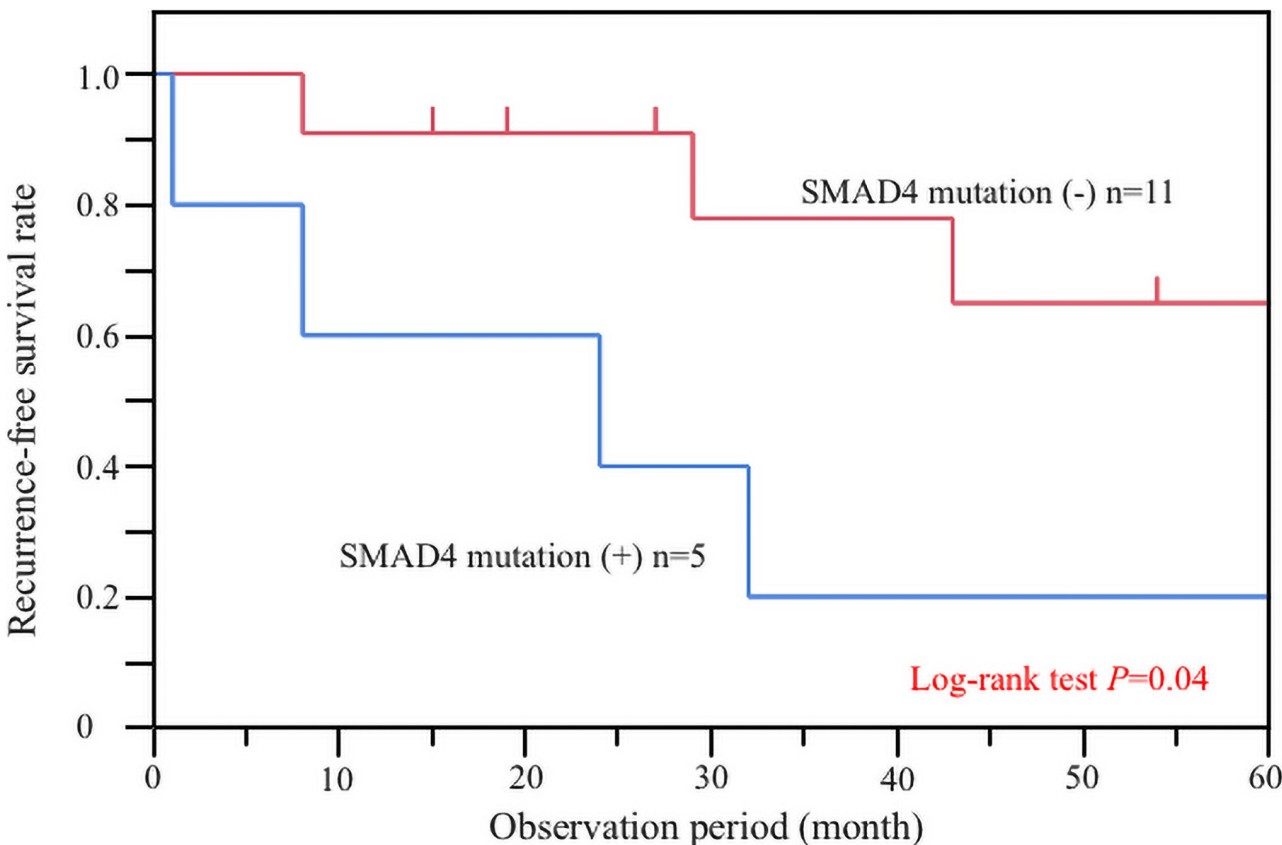

**Fig 3. Analysis of recurrence-free survival (RFS) in patients with small-bowel cancer related to *SMAD4* mutation.** Patients with mutant *SMAD4* had significantly poorer prognosis than those with wild-type *SMAD4* (*P* = 0.04).

**Table 6. Univariate association with recurrence-free survival.**

| Variables | | N | RFS | | |
|---|---|---|---|---|---|
| | | | HR (95%CI) | *P*-value | Log-rank *P*-value |
| Sex | Male | 10 | 2.98 (0.50–56.65) | 0.26 | 0.29 |
| | Female | 6 | Ref. | | |
| Age | >65 years | 7 | 1.21 (0.23–5.61) | 0.81 | 0.81 |
| | <65 years | 9 | Ref. | | |
| Stage | IV | 3 | 11.43 (1.09–247.04) | 0.04 | 0.01 |
| | I–III | 13 | Ref. | | |
| Postoperative chemotherapy | Done | 10 | 1.74 (0.38–12.22) | 0.49 | 0.50 |
| | None | 6 | Ref. | | |
| IHC | dMMR | 8 | 0.69 (0.14–3.13) | 0.62 | 0.62 |
| | pMMR | 8 | Ref. | | |
| TMB | <10 mut/Mb | 3 | 2.97 (0.41–15.37) | 0.25 | 0.18 |
| | ≥10 mut/Mb | 13 | Ref. | | |
| *SMAD4* | Mutation | 5 | 4.15 (0.90–21.41) | 0.07 | 0.04 |
| | Wild | 11 | Ref. | | |

IHC: immunohistochemical, TMB: tumor mutational burden, mut/MB: mutations per megabase of DNA, MMR: mismatch repair, dMMR: deficient mismatch repair, pMMR: proficient mismatch repair, HR: hazard ratio, RFS: recurrence-free survival.

**Table 7. Multivariate association with recurrence-free survival.**

| Variables | | N | RFS | |
|---|---|---|---|---|
| | | | HR (95%CI) | *P*-value |
| Stage | IV | 3 | 24.07 (1.84–654.62) | 0.02 |
| | I–III | 13 | Ref. | |
| *SMAD4* | Mutation | 5 | 6.72 (1.28–49.54) | 0.03 |
| | Wild | 11 | Ref. | |

HR: hazard ratio, RFS: recurrence-free survival.

esophagogastroduodenoscopy, whereas only 11% of patients with small-bowel cancer were diagnosed with early-stage disease [4]. A recent study on genomic variants demonstrated that *CDKN2A/B* and *ERBB2/HER2* variants are more enriched in duodenal cancer than in small-bowel cancer and duodenal cancer overall tends to have lower TMB [13].

First, our results showed that the genes commonly altered in small-bowel cancer in our Japanese population were *TP53*, *KRAS*, *ARID1A*, *PIK3CA*, *APC*, *SMAD4*, *NOTCH3*, *CREBBP*, *PTCH1*, and *EP300*. In previous reports from Western countries, the most commonly mutated genes identified in small-bowel cancer include *TP53* (41%–58%), *KRAS* (27%–54%), *APC* (13%–27%), *SMAD4* (10%–17%), and *PIK3CA* (9%–16%) [11, 13–15]. The frequencies of these genomic variants are similar to our results; however, the frequencies of *APC* mutations in those studies are higher than reported previously [11, 13–15] (S4 Fig). It has been reported that adenoma-carcinoma sequence usually does not result in the development of small-bowel cancers, instead the low frequency of *APC* gene mutations is responsible for their development [18]. Hänninen et al. [14] reported that small-bowel cancers in 9.4% of the patients are associated with celiac disease. Furthermore, Diosdado et al. [19] reported that *APC* promoter methylation is a common event in celiac disease-related small-bowel cancers. As *APC* mutations occur exclusively in small-bowel cancer patients without IBD and celiac disease, this may strongly affect our findings as no patients of IBD or celiac disease were observed in our study.

Second, TMB status was strongly associated with prognosis in patients with small-bowel cancer. TMB is a measurement of somatic mutations per Mb of DNA carried by tumor cells. Standard values of high TMB are generally determined on an organ-to-organ basis [20]; however, this has not been defined for small-bowel cancer due to its rare occurrence. In our study, the frequency of small-bowel cancer patients with TMB ≥10 mutations/Mb was 66.7%, compared to 12.3% in a previous study [13]. This may depend on the inclusion of duodenal cancer samples, as TMB was significantly higher in small-bowel cancers compared to that in duodenal cancers in a previous study [13]. Additionally, elevated neoantigen load is associated with improved CRC-specific survival via lymphocyte infiltration in the tumor microenvironment [21], and patients with a high TMB have better prognosis than those with a low TMB [22]. This phenomenon observed in CRC is also considered to occur in small-bowel cancer. Therefore, small-bowel cancers with high TMB are believed to express mutation-associated neoantigens. In addition, immune checkpoint inhibitors effective at treating metastatic CRC with high TMB have been recently developed [23]. These immune checkpoint inhibitors may also be effective in treating small-bowel cancers with high TMB. Although, none of the patients in our current study were treated with immune checkpoint inhibitors, the results of the CRC study suggest that the TMB status may be an important biomarker for the prognosis of small-bowel cancer.

Finally, the RFS of patients with small-bowel cancers containing *SMAD4* mutations was significantly poorer than that of patients without *SMAD4* mutations. SMAD proteins are key

signal transducers of the transforming growth factor (TGF)-β signaling pathway, which plays a critical role in tumor progression [24]. *SMAD4* is localized at the chromosome band 18q21; it functions as a tumor suppressor in the TGF-β signaling pathway, thereby regulating cell proliferation, differentiation, morphogenesis, and apoptosis [25]. The loss of *SMAD4* function is an independent prognostic factor for decreased RFS and OS in patients with CRC [26–30]. However, there are currently no reports regarding the significance of *SMAD4* variants in patients with small-bowel cancer. Two of the 3 patients diagnosed at Stage II with *SMAD4* mutation had recurrence by peritoneal dissemination. Currently, adjuvant chemotherapy is not recommended for all patients with Stage II CRC in Japan; however, these results suggest that postoperative adjuvant chemotherapy should be considered for Stage II small-bowel cancer with *SMAD4* mutations because of the high risk of recurrence. Although no correlation was observed between DSS and OS, possibly because of the small number of patients analyzed, further studies may indicate *SMAD4* mutations as a prognostic factor for small-bowel cancer, as in CRC.

Our study had several limitations. First, this study was a retrospective study conducted at two regional centers that may have resulted in recruitment bias. Second, the number of participants was small. Third, our study included only surgical resection patients as we excluded patients treated with endoscopic resection or chemotherapy. Finally, we used a target panel with only cancer-related genes. Therefore, it is possible that some genes important for the development of small-bowel cancers were eliminated from the whole genomic sequencing. However, while most previous studies included duodenal cancer, our study specifically focused on rare small-bowel cancer. Even though this contributed to a small sample size, the outcomes of this pilot study will help in genomic characterization of small-bowel cancer in large prospective cohort studies in the future.

In conclusion, TMB levels correlated with tumor prognosis and *SMAD4* mutations were associated with recurrence after R0 resection in patients with small-bowel cancer. Our results provide novel insights for the development of novel therapeutic approaches such as molecularly targeted drugs for small-bowel cancers.

## Supporting information

**S1 Fig. Analysis of overall survival (OS) and disease-specific survival (DSS) related to mismatch repair (MMR) status.** (**a**) OS of patients with small-bowel cancer related to MMR status. (**b**) DSS of patients with small-bowel cancer related to the MMR status. There were no significant differences in survival associated with MMR status.
(TIFF)

**S2 Fig. Analysis for overall survival (OS) and disease-specific survival (DSS) related to each gene mutation.** (**a**) OS of patients with small-bowel cancer and mutations in *TP53* (A), *KRAS* (B), *PIK3CA* (C), and *SMAD4* (D). (**b**) DSS of patients with small-bowel cancer and mutations in *TP53* (A), *KRAS* (B), *PIK3CA* (C), and *SMAD4* (D). There were no significant differences in survival associated with any gene mutation.
(TIFF)

**S3 Fig. Analysis of recurrence-free survival (RFS) related to each gene mutation.** RFS of patients with small-bowel cancer related to tumor mutational burden (TMB; A) and mutations in *KRAS* (B), *TP53* (C), and *PIK3CA* (D). There were no significant differences in RFS associated with TMB or any gene mutation.
(TIFF)

**S4 Fig. Comparison of common mutated gene frequency of small-bowel cancers between the current study and previous studies.** We compared the frequency of genomic variants in *TP53*, *KRAS*, *APC*, *SMAD4*, and *PIK3CA* for small-bowel cancers. There were no significantly different genomic variants for any gene. However, the frequency of genomic variant of *APC* and *PIK3CA* in this study tended to be higher than that reported in previous studies. (TIFF)

**S1 Table. Target panel of 90 cancer-related genes used in our study.**
(DOCX)

**S2 Table. Outcome of immunohistochemistry analysis.**
(DOCX)

**S3 Table. Characteristics of lesions with high TMB and low TMB.**
(DOCX)

## Author Contributions

**Writing – original draft:** Akiyoshi Tsuboi.

**Writing – review & editing:** Yuji Urabe, Shiro Oka, Akihiko Sumioka, Sumio Iio, Ryo Yuge, Ryohei Hayashi, Toshio Kuwai, Yasuhiko Kitadai, Kazuya Kuraoka, Koji Arihiro, Shinji Tanaka, Kazuaki Chayama.

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
