## [Decision Letter · Decision Letter 0]

9 Sep 2020

PONE-D-20-24291

Genomic analysis for the prediction of prognosis in small-bowel cancer

PLOS ONE

Dear Dr. Oka,

Thank you for submitting your manuscript to PLOS ONE. After careful consideration, we feel that it has merit but does not fully meet PLOS ONE’s publication criteria as it currently stands. Therefore, we invite you to submit a revised version of the manuscript that addresses the points raised during the review process.

In particular, please address the statistical analysis is necessary to elucidate the contributions of TMB and SMAD4 over known prognostic factors such as R0 resection rate, stage, etc. using a Cox proportional hazards model or similar approach. Also, note the other issues rasied by the reviewers, such at the high rate of mismatch repair deficiency.

We look forward to receiving your revised manuscript.

Kind regards,

Anthony F. Shields, M.D., Ph.D.

Academic Editor

PLOS ONE

2. In ethics statement in the manuscript and in the online submission form, please provide additional information about the patient records used in your retrospective study. Specifically, please ensure that you have discussed whether all data were fully anonymized before you accessed them and/or whether the IRB or ethics committee waived the requirement for informed consent for the use of medical data and tissue samples. If patients provided informed written consent to have data from their medical records and tissues used in research, please include this information.

Reviewers' comments:

**Comments to the Author**

1. Is the manuscript technically sound, and do the data support the conclusions?

Reviewer #1: Yes

Reviewer #2: Yes

2. Has the statistical analysis been performed appropriately and rigorously? 

Reviewer #1: No

Reviewer #2: Yes

3. Have the authors made all data underlying the findings in their manuscript fully available?

Reviewer #1: Yes

Reviewer #2: Yes

4. Is the manuscript presented in an intelligible fashion and written in standard English?

Reviewer #1: Yes

Reviewer #2: Yes

5. Review Comments to the Author

Reviewer #1: The authors present a genomic analysis of small bowel cancer in a retrospective study. The findings indicating that TMB-high patients had improved survival is notable; however, many other conclusions are limited by the small sample size of 24 patients, all of whom underwent surgical resection and did not include duodenal cancers.

-If data on patient treatment (e.g. adjuvant therapy) were included, it would strengthen the novelty of the manuscript.

-The genetics were similar to Western data, why do the offers suggest that small bowel cancer has a different biology in the Japanese population?

-A more rigorous statistical analysis is necessary to elucidate the contributions of TMB and SMAD4 over known prognostic factors such as R0 resection rate, stage, etc. using a Cox proportional hazards model.

-Breast is misspelled in Table 1.

Reviewer #2: This is a well-written manuscript on a very interesting topic in a highly selected population of small-bowel cancer (duodenum not included).

The main issue here is the number of patients included in the analysis, which make results quite "weak" and not robust enough to be reliable.

Major:

-TMB cut off: please clarify why 10 is used as cutoff.

-MMR deficiency rates are higher then expected, sure this may have an impact both in TMB and in OS data. Please clarify this in the manuscript.

-Au say "Although, we did not observe any correlation between DSS and OS, which may be due

to the small number of cases analyzed, we believe that the SMAD4 mutations may be a

prognostic factor for small-bowel cancer, similar to that of CRC. " Please provide more data and explanations for this statement.

---

## [Author Response · Author response to Decision Letter 0]

6 Oct 2020

Reviewer #1: The authors present a genomic analysis of small bowel cancer in a retrospective study. The findings indicating that TMB-high patients had improved survival is notable; however, many other conclusions are limited by the small sample size of 24 patients, all of whom underwent surgical resection and did not include duodenal cancers.

-If data on patient treatment (e.g. adjuvant therapy) were included, it would strengthen the novelty of the manuscript.

Reply: Thank you for pointing out this matter. We have included adjuvant therapy in Table 1a under ‘Diagnostic modality’ as follows:

“Table 1 Baseline characteristics of small-bowel cancer

1a. Characteristics patients with small-bowel cancer in our study

Variables Small-bowel cancer (n = 24)

Age (years), mean ± SD 61.7 ± 11.7

Sex, male/female 16/8

BMI, mean ± SD 22.3 ± 6.3

Observation period (month), mean ± SD 50.0 ± 43.3

Chief complaint, +/− 22/2

Intestinal obstruction 9 (38)

Abdominal pain 8 (33)

Gastrointestinal bleeding 5 (21)

History of other organ cancers 7 (29)

Colorectal cancer 3(13)

Breast cancer 2(8)

Gastric cancer 1(4)

Lung cancer 1(4)

Bladder cancer 1(4)

Pancreatic cancer 1(4)

Endometrial cancer 1(4)

Gallbladder cancer 1(4)

Predisposing conditions 

FAP 0 (0)

PJS 0 (0)

LS 2 (8)

Crohn’s disease 0 (0)

Celiac disease 0 (0)

Number of lesions, single/ multiple 21/3

CEA, median [IQR] 2.2 ng/mL [1.4–5.9]

CA19-9, median [IQR] 8.5 U/mL [2–77.9]

Diagnostic modality 

DBE 20 (83)

CT 3 (13)

CE 1 (4)

Adjuvant therapy 

TS1 5 (21)

FOLFOX 4 (17)

CapeOX 3 (13)

Capecitabine 2 (8)

UFT/LV 2 (8)

FOLFOX+Bmab 1 (4)

TS1+Doc 1 (4)

None 6 (25)

Data represented as n (%) and mean ± SD.

SD: standard deviation, BMI: body mass index, FAP: familial adenomatous polyposis, PJS: Peutz-Jeghers syndrome, LS: Lynch syndrome, CEA: carcinoembryonic antigen, CA19-19: carbohydrate antigen 19-9, IQR: interquartile range, DBE: double balloon endoscopy, CT: computed tomography, CE: capsule endoscopy”

A total of 18 of the 24 patients received postoperative adjuvant chemotherapy and the regimens were as follows: 5 patients received TS-1, 4 patients received FOLFOX, 3 patients received CapeOX, 2 patients received Capecitabine, 2 patients received UFT/LV, 1 patient received FOLFOX+B-mab, and 1 patient received TS-1+ Docetaxel. Postoperative adjuvant chemotherapy was not correlated with OS (HR, 2.47; 95% CI, 0.64–16.16; P = 0.20) (Table 3). All patients (6/6) with TMB <10 mut/Mb received postoperative adjuvant chemotherapy, while 65% of the patients (11/17) with TMB ≥10 mut/Mb received postoperative adjuvant chemotherapy. All patients (6/6) with TMB <10 mut/Mb that received postoperative adjuvant chemotherapy had a worse prognosis than those with TMB ≥10 mut/Mb. Therefore, we believe that cancer genomic analysis of resected specimens may allow for indication of postoperative adjuvant chemotherapy depending on the value of TMB.

-The genetics were similar to Western data, why do the offers suggest that small bowel cancer has a different biology in the Japanese population?

Reply: We probably misled you when expressing our opinion without giving it much thought. The frequencies of redisposing factors of small-bowel cancer such as FAP, LS, Crohn’s disease, celiac disease, and BMI are different in Japanese and Caucasian populations. Thus, we suggest that these factors may have an effect on the development of small-bowel cancer. The genetics were indeed similar to that of Western data; however, some difference was found in the frequencies of APC mutations between our data and previous reports from Western countries. 

We have altered some sentences in the revised version of the paper as follows:

(p.5 lines 81–88)

From “However, significantly different factors are involved the development of small-bowel cancer exist between Caucasian and Japanese populations.” 

To “However, disease structure is different between Caucasian and Japanese populations. For example, background factors such as the frequency of celiac disease, Crohn's disease, and obesity are different between Japan and Western countries. The frequency of obesity varies across countries. According to World Health Organization (2016), a 20–40% frequency of body mass index (BMI) >30 was reported in Western countries, whereas a 4.4% frequency of BMI >30 was reported in Japan; this scenario is explained by Japanese people tending to consume diets low in fat, sugar and fructose.”

-A more rigorous statistical analysis is necessary to elucidate the contributions of TMB and SMAD4 over known prognostic factors such as R0 resection rate, stage, etc. using a Cox proportional hazards model.

Reply: Thank you for pointing this out. After re-examining the Cox proportional hazards model, we found that Stage and TMB were correlated with OS, and that Stage and SMAD4 were correlated with RFS. These results suggest that the presence of SMAD4 mutations and TMB status are strongly correlated. These results are presented in Tables 3–6) as follows:

(p. 17–19)

Table 3 Univariate association with overall survival

Variables N OS

 HR (95%CI) P-value Log-rank

P-value

Sex Male 16 2.60 (0.68–16.94) 0.17 0.20

 Female 8 Ref. 

Age >65 8 0.86 (0.23–2.73) 0.80 0.80

 ≦65 16 Ref. 

pStage IV 9 12.13 (2.89–83.36) <0.01 <0.01

 I–III 15 Ref. 

R0 resection R1/2 8 7.7 (2.06–37.25) <0.01 <0.01

 R0 16 Ref. 

Adjuvant chemotherapy Done 18 2.47 (0.64–16.16) 0.20 0.23

 None 6 Ref. 

IHC dMMR 11 1.25 (0.38–3.99) 0.71 0.71

 pMMR 13 Ref. 

TMB <10 6 3.29 (0.84–11.24) 0.08 <0.05

 ≧10 17 Ref. 

SMAD4 Mutation 6 0.52 (0.08–2.01) 0.37 0.39

 Wild 17 Ref. 

Data represented as n.

IHC: immunohistochemical, TMB: tumor mutational burden, mut/MB: mutations per megabyte of DNA, MMR: mismatch repair, dMMR: deficient mismatch repair, pMMR: proficient mismatch repair, HR: hazard ration, OS: overall survival

(p. 18)

Table 4 Multivariate association with overall survival

Variables N OS

 HR (95%CI) P-value

pStage IV 9 58.68 (7.89–1348.14) <0.01

 I–III 15 Ref. 

TMB <10 6 11.33 (2.08–85.05) <0.01

 ≧10 17 Ref. 

Data represented as n.

TMB: tumor mutational burden, mut/MB: mutations per megabyte of DNA, HR: hazard ration, OS: overall survival

(p. 20)

Table 5 Univariate association with recurrence-free survival

Variables N RFS

 HR (95%CI) P-value Log-rank

P-value

Sex Male 10 2.98 (0.50–56.65) 0.26 0.29

 Female 6 Ref. 

Age >65 7 1.21 (0.23–5.61) 0.81 0.81

 ≦65 9 Ref. 

pStage IV 3 11.43 (1.09–247.04) 0.04 0.01

 I–III 13 Ref. 

Adjuvant chemotherapy Done 10 1.74 (0.38–12.22) 0.49 0.50

 None 6 Ref. 

IHC dMMR 8 0.69 (0.14–3.13) 0.62 0.62

 pMMR 8 Ref. 

TMB <10 3 2.97 (0.41–15.37) 0.25 0.18

 ≧10 13 Ref. 

SMAD4 Mutation 5 4.15 (0.90–21.41) 0.07 0.04

 Wild 11 Ref. 

Data represented as n.

IHC: immunohistochemical, TMB: tumor mutational burden, mut/MB: mutations per megabyte of DNA, MMR: mismatch repair, dMMR: deficient mismatch repair, pMMR: proficient mismatch repair, HR: hazard ration, OS: overall survival

(p. 20–21)

Table 6 Multivariate association with recurrence-free survival

Variables N OS

 HR (95%CI) P-value

pStage IV 3 24.07 (1.84–654.62) 0.02

 I–III 13 Ref. 

TMB <10 5 6.72 (1.28–49.54) 0.03

 ≧10 11 Ref. 

Data represented as n.

TMB: tumor mutational burden, mut/MB: mutations per megabyte of DNA, HR: hazard ration, RFS: recurrence-free survival

Moreover, we included the following text in the Results section:

(p. 17 lines 290–297)

“The other covariates from the Cox proportional hazard analysis associated with OS are shown in Tables 3 and 4. Regarding the univariate analysis, pathological Stage IV (HR, 12.13; 95% CI, 2.89–83.36; P < 0.01) and R1/2 resection (HR, 7.7; 95% CI, 2.06–37.25; P < 0.01) were correlated with OS (Table 3), whereas TMB ≥10 mutations/Mb was not significantly correlated with OS (HR, 3.29; 95% CI, 0.84–11.24; P = 0.08). However, in the multivariate analysis, pathological Stage IV (HR, 58.68; 95% CI, 7.89–1348.14; P < 0.01) and TMB ≥10 mutations/Mb (HR, 11.33; 95% CI, 2.08–85.05; P < 0.01) were correlated with OS.”

(p. 19 lines 315–321)

“The other covariates from the Cox proportional hazard analysis associated with RFS are shown in Tables 5 and 6. Regarding the univariate analysis, pathological Stage IV (HR, 11.43; 95% CI, 1.09–247.04; P = 0.04) was correlated with RFS (Table 5), whereas mutated SMAD4 was not significantly correlated with RFS (HR, 4.15; 95% CI, 0.90–21.41; P = 0.07). However, pathological Stage IV (HR, 24.07; 95% CI, 1.84–654.62; P = 0.02) and mutated SMAD4 (HR, 6.72; 95% CI, 1.28–49.54; P = 0.03) were correlated with RFS in the multivariate analysis.”

-Breast is misspelled in Table 1.

Reply: Thank you for pointing this out. I corrected the spelling in Table 1.

Reviewer #2: This is a well-written manuscript on a very interesting topic in a highly selected population of small-bowel cancer (duodenum not included).

The main issue here is the number of patients included in the analysis, which make results quite "weak" and not robust enough to be reliable.

Major:

-TMB cut off: please clarify why 10 is used as cutoff.

Reply: Thank you for pointing that out. The problem here is that the cut-off value of TMB for small-bowel cancer has not yet been determined. However, Shrock et al. (JAMA Oncol. 2017; 3: 1546-1553.) defined the TMB status of <10 as low, 10–20 as intermediate, and 20> as high. Therefore, in the present study, we defined TMB ≥10 mutations/Mb as being a high value. 

-MMR deficiency rates are higher then expected, sure this may have an impact both in TMB and in OS data. Please clarify this in the manuscript.

Reply: Thank you for pointing this out. The Cox proportional hazards model showed that dMMR was not correlated with OS, but the stepwise method showed that Stage and TMB were correlated with OS. We describe this in the Results section: 

(p. 17 lines 290–297)

“The other covariates from the Cox proportional hazard analysis associated with OS are shown in Tables 3 and 4. Regarding the univariate analysis, pathological Stage IV (HR, 12.13; 95% CI, 2.89–83.36; P < 0.01) and R1/2 resection (HR, 7.7; 95% CI, 2.06–37.25; P < 0.01) were correlated with OS (Table 3), whereas TMB ≥10 mutations/Mb was not significantly correlated with OS (HR, 3.29; 95% CI, 0.84–11.24; P = 0.08). However, in the multivariate analysis, pathological Stage IV (HR, 58.68; 95% CI, 7.89–1348.14; P < 0.01) and TMB ≥10 mutations/Mb (HR, 11.33; 95% CI, 2.08–85.05; P < 0.01) were correlated with OS.”

(p. 17–19)

Table 3 Univariate association with overall survival

Variables N OS

 HR (95%CI) P-value Log-rank

P-value

Sex Male 16 2.60 (0.68–16.94) 0.17 0.20

 Female 8 Ref. 

Age >65 8 0.86 (0.23–2.73) 0.80 0.80

 ≦65 16 Ref. 

pStage IV 9 12.13 (2.89–83.36) <0.01 <0.01

 I–III 15 Ref. 

R0 resection R1/2 8 7.7 (2.06–37.25) <0.01 <0.01

 R0 16 Ref. 

Adjuvant chemotherapy Done 18 2.47 (0.64–16.16) 0.20 0.23

 None 6 Ref. 

IHC dMMR 11 1.25 (0.38–3.99) 0.71 0.71

 pMMR 13 Ref. 

TMB <10 6 3.29 (0.84–11.24) 0.08 <0.05

 ≧10 17 Ref. 

SMAD4 Mutation 6 0.52 (0.08–2.01) 0.37 0.39

 Wild 17 Ref. 

Data represented as n.

IHC: immunohistochemical, TMB: tumor mutational burden, mut/MB: mutations per megabyte of DNA, MMR: mismatch repair, dMMR: deficient mismatch repair, pMMR: proficient mismatch repair, HR: hazard ration, OS: overall survival

(p. 18)

Table 4 Multivariate association with overall survival

Variables N OS

 HR (95%CI) P-value

pStage IV 9 58.68 (7.89–1348.14) <0.01

 I–III 15 Ref. 

TMB <10 6 11.33 (2.08–85.05) <0.01

 ≧10 17 Ref. 

Data represented as n.

TMB: tumor mutational burden, mut/MB: mutations per megabyte of DNA, HR: hazard ration, OS: overall survival

-Au say "Although, we did not observe any correlation between DSS and OS, which may be due to the small number of cases analyzed, we believe that the SMAD4 mutations may be a prognostic factor for small-bowel cancer, similar to that of CRC. " Please provide more data and explanations for this statement.

Reply: Thank you for pointing this out. The case of recurrence with SMAD4 mutation is relatively recent, and the patient is still undergoing multidisciplinary treatment. However, there are no current cases of CR, which may affect OS and DSS in the future as we perform the long-term follow up. More detailed data are provided in the Results section:

(p. 19 lines 324–329)

“Of the four patients that had recurrence of small-bowel cancer, two patients died from primary cancer, whereas the other two patients are still alive and undergoing treatment with multidisciplinary treatment including additional surgery. Of the two patients that had recurrence at Stage II, one of them received postoperative adjuvant chemotherapy with capecitabine, and the other did not receive adjuvant chemotherapy.”

and in the Discussion section:

(p. 25 lines 412–419)

“Two of the three patients diagnosed at Stage II with SMAD4 mutation had recurrence by peritoneal dissemination. Currently, adjuvant chemotherapy is not recommended for all patients with Stage II CRC in Japan; however, these results suggest that postoperative adjuvant chemotherapy should be considered for Stage II small-bowel cancer with SMAD4 mutations because of the high risk of recurrence. Although no correlation was observed between DSS and OS, possibly because of the small number of cases analyzed, further studies may indicate SMAD4 mutations as a prognostic factor for small-bowel cancer, as in CRC.”

---

## [Editor Report · Decision Letter 1]

15 Oct 2020

Genomic analysis for the prediction of prognosis in small-bowel cancer

PONE-D-20-24291R1

Dear Dr. Oka,

We’re pleased to inform you that your manuscript has been judged scientifically suitable for publication and will be formally accepted for publication once it meets all outstanding technical requirements.

Kind regards,

Anthony F. Shields, M.D., Ph.D.

Academic Editor

PLOS ONE

Additional Editor Comments (optional):

In Table 1 please define the abbreviations used for the adjuvant treatment regimens.

---

## [Editor Report · Acceptance letter]

11 May 2021

PONE-D-20-24291R1 

Genomic analysis for the prediction of prognosis in small-bowel cancer 

Dear Dr. Oka:

I'm pleased to inform you that your manuscript has been deemed suitable for publication in PLOS ONE. Congratulations! Your manuscript is now with our production department. 

Kind regards, 

on behalf of

Dr. Anthony F. Shields 

Academic Editor

PLOS ONE